# Clues for Improving the Pathophysiology Knowledge for Endometriosis Using Plasma Micro-RNA Expression

**DOI:** 10.3390/diagnostics12010175

**Published:** 2022-01-12

**Authors:** Yohann Dabi, Stéphane Suisse, Ludmila Jornea, Delphine Bouteiller, Cyril Touboul, Anne Puchar, Emile Daraï, Sofiane Bendifallah

**Affiliations:** 1Department of Obstetrics and Reproductive Medicine, Hôpital Tenon, Sorbonne University, 4 Rue de la Chine, 75020 Paris, France; yohann.dabi@gmail.com (Y.D.); cyril.touboul@gmail.com (C.T.); anne.puchar@aphp.fr (A.P.); emile.darai@aphp.fr (E.D.); 2Clinical Research Group (GRC) Paris 6, Centre Expert Endométriose (C3E), Sorbonne University (GRC6 C3E SU), 4 Rue de la Chine, 75020 Paris, France; 3Cancer Biology and Therapeutics, Centre de Recherche Saint-Antoine (CRSA), Sorbonne University, INSERM UMR_S_938, 75020 Paris, France; 4Ziwig Health, 19 Rue Reboud, 69003 Lyon, France; stephane@ziwig.com; 5Paris Brain Institute—Institut du Cerveau—ICM, Inserm U1127, CNRS UMR 7225, AP-HP—Hôpital Pitié-Salpêtrière, Sorbonne University, 4 Rue de la Chine, 75020 Paris, France; ludmila.jornea@icm-institute.org; 6Gentoyping and Sequencing Core Facility, iGenSeq, Institut du Cerveau et de la Moelle Épinière, ICM, Hôpital Pitié-Salpêtrière, 47-83 Boulevard de l’Hôpital, 75013 Paris, France; delphine.bouteiller@icm-institute.org

**Keywords:** endometriosis, miRNA, pathophysiology, pathways

## Abstract

The pathophysiology of endometriosis remains poorly understood. The aim of the present study was to investigate functions and pathways associated with the various miRNAs differentially expressed in patients with endometriosis. Plasma samples of the 200 patients from the prospective “ENDO-miRNA” study were analyzed and all known human miRNAs were sequenced. For each miRNA, sensitivity, specificity, and ROC AUC values were calculated for the diagnosis of endometriosis. miRNAs with an AUC ≥ 0.6 were selected for further analysis. A comprehensive review of recent articles from the PubMed, Clinical Trials.gov, Cochrane Library, and Web of Science databases was performed to identify functions and pathways associated with the selected miRNAs. In total, 2633 miRNAs were found in the patients with endometriosis. Among the 57 miRNAs with an AUC ≥ 0.6: 20 had never been reported before; one (miR-124-3p) had previously been observed in endometriosis; and the remaining 36 had been reported in benign and malignant disorders. miR-124-3p is involved in ectopic endometrial cell proliferation and invasion and plays a role in the following pathways: mTOR, STAT3, PI3K/Akt, NF-κB, ERK, PLGF-ROS, FGF2-FGFR, MAPK, GSK3B/*β*–catenin. Most of the remaining 36 miRNAs are involved in carcinogenesis through cell proliferation, apoptosis, and invasion. The three main pathways involved are Wnt/*β*–catenin, PI3K/Akt, and NF–KB. Our results provide evidence of the relation between the miRNA profiles of patients with endometriosis and various signaling pathways implicated in its pathophysiology.

## 1. Introduction

Endometriosis, defined by the presence of endometrial-like tissue outside the uterus, affects 5–10% of women of reproductive age, but is also diagnosed in menopausal patients with an incidence estimated at 2–5% [1,2]. In the premenopausal period, diagnosis is mainly based on symptoms including severe chronic pelvic pain, dysmenorrhea, dyspareunia, dyschezia, and infertility. However, no single sign is sufficiently characteristic to make a diagnosis. In postmenopausal patients, endometriosis can be symptomatic but is also diagnosed in a context associated with, or mimicking, a cancer. From the pathophysiology point of view, endometriosis is considered a multifactorial disease with genetic and epigenetic controls involving multiple pathways such as cell proliferation, cell differentiation, cell adhesion, apoptosis, angiogenesis, steroidogenesis, inflammatory and immune responses, oncogenic suppressors, as well as exposome factors, particularly persistent organic pollutants (POP) [3]. However, despite numerous investigations of these various pathways, the pathophysiology of endometriosis remains an enigma.

Human miRNAs are single-stranded highly conserved non-coding RNAs composed of 21–25 nucleotides binding to their complementary mRNAs regulating their degradation and translation [4,5]. It is estimated that about 60% of genes are regulated by miRNAs [6,7]. Cumulative evidence suggests that miRNA dysregulation plays a pivotal role in many benign and malignant disorders, as well as in endometriosis which shares features of both pathologies. More than 2600 miRNAs have been identified in humans to date. Evaluation of these miRNAs in patients with endometriosis has shown that more than 200 are differentially expressed in patients with and without endometriosis [8,9,10] with some literature on their relevance in endometriosis’ diagnostic [11,12,13]. Among these 200 miRNAs, only a few have been analyzed with a view to better understanding the pathophysiology of endometriosis [14,15].

Therefore, using data from the prospective “ENDOmiRNA” study [16], the aim of the present work was to investigate functions and pathways associated with the various miRNAs differentially expressed in patients with and without endometriosis to highlight new potential fields for research and treatments.

## 2. Materials and Methods

### 2.1. Study Population

We used data from the prospective “ENDOmiARN” study (ClinicalTrials.gov Identifier: NCT04728152). Data collection and analysis were carried out under the Research Protocol n° ID RCB: 2020-A03297-32. The ENDOmiARN study included 200 plasma samples obtained from patients with chronic pelvic pain suggestive of endometriosis. All had undergone a laparoscopic procedure (either therapeutic or diagnostic laparoscopic) and/or MRI imaging evidencing endometriosis by the presence of endometrioma and/or deep endometriosis [17,18,19], as stated in the trial registration. The laparoscopy procedures were systematically recorded and the video analyzed by two operators (CT, YD), who were blind to the symptoms and imaging findings, to confirm the presence or absence of endometriosis. All patients undergoing diagnostic or operative laparoscopy underwent a systematic histological confirmation of endometriosis when potential lesions were present. For the patients in the endometriosis group without laparoscopic evaluation, all had MRI features of deep endometriosis with colorectal involvement and/or endometrioma have been revised in the multidisciplinary endometriosis committee. All the plasma samples were collected between January 2021 and June 2021. All samples were collected at the first consultation, prior to laparoscopy (in patients that underwent surgery). Analysis was performed blinded to the surgical and imaging findings. If endometriosis was detected, the subjects were stratified according to the revised American Society of Reproductive Medicine (rASRM) classification [20].

### 2.2. Plasma Sample Collection

Blood samples (4 mL) were collected in EDTA tubes (BD, Franklin Lakes, NJ, USA). The plasma was then isolated from whole blood within a maximum of 2 h by two successive centrifugations at 4 °C (first at 1900× *g* (3000 rpm) for 10 min, followed by 13,000–14,000× *g* for 10 min to remove all cell debris), and aliquoted, labeled, and stored at −80 °C until analysis, as previously published [21,22,23].

### 2.3. RNA Sample Extraction, Preparation, and Quality Control

The RNA was extracted from 500 μL of plasma on a Maxwell 48^®^ RSC automat using the Maxwell^®^ RSC miRNA Plasma and Serum Kit (ref AS1680, Promega, Madison, WI, USA) according to the manufacturer’s protocol. Libraries for small RNA sequencing were prepared using the QIAseq miRNA Library Kit for Illumina (Qiagen, Hilden, Germany). The resulting small RNA libraries were concentrated by ethanol precipitation and quantified using a Qubit 2.0 Fluorometer (Thermo Fisher Scientific, Waltham, MA, USA) prior to sequencing on a Novaseq 6000 sequencer (Illumina, San Diego, CA, USA) with read lengths of 100 bases and 17 million single-end reads per sample, on average [24,25].

## 3. Bioinformatics

### 3.1. Raw Data Preprocessing (Raw, Filtered, Aligned Reads) and Quality Control

Sequencing reads were processed using the data processing pipeline. FastQ files were trimmed to remove adapter sequences using Cutadapt version v.1.18 and were aligned using Bowtie version 1.1.1 to the following transcriptome databases: the human reference genome available from NCBI (https://www.ncbi.nlm.nih.gov/genome/guide/human/, accessed on 28 August 2021), and miRBase22 (miRNAs) using the MirDeep2 v0.1.0 package. The raw sequencing data quality was assessed using FastQC software v0.11.7 [25,26,27,28,29]. The bioinformatic process used was previously described by Potla et al. [30].

### 3.2. Differential Expression Analysis of the miRNAs

miRNA expression was quantified by miRDeep2 [31]. Differential expression tests were then conducted in DESeq2 for miRNAs with read counts in ≥1 of the samples. DESeq2 integrates methodological advances with several novel features to facilitate more quantitative analysis of comparative RNA-seq data using shrinkage estimators for dispersion and fold change [32,33]. miRNAs were considered as differentially expressed if the absolute value of log2-fold change was >1.5 (upregulated) and <0.5 (downregulated). The *p*-value adjusted for multiple testing was <0.05 [32].

### 3.3. Study of the miRNA Accuracy

To evaluate the diagnostic accuracy of each miRNA biomarker, sensitivity, specificity, and ROC analysis was performed, and the ROC AUC was calculated [34,35].

Additional statistical analysis was based on the Chi2 test as appropriate for categorical variables. Values of *p* < 0.05 were considered to denote significant differences. Data were managed with an Excel database (Microsoft, Redmond, WA, USA) and analyzed using R 2.15 software, available online (http://cran.r-project.org/, accessed on 28 August 2021).

### 3.4. Sources and Search Strategy

The PubMed, ClinicalTrials.gov, Cochrane Library, and Web of Science databases were queried for relevant studies published before 1 July 2021 using the miRNAs names exclusively as search terms. All English results were screened to perform a comprehensive evaluation of relevant articles.

## 4. Results

### 4.1. Demographic Characteristics of the Population

The ENDOmiARN study included 200 patients, 76.5% (*n* = 153) and 23.5% (*n* = 47) have been diagnose with and without endometriosis, respectively. Clinical characteristics of the endometriosis and controls patients are displayed in Table 1. None of the patients had a history of ovarian cancer in the cohort. Within the group of patients diagnosed with endometriosis, a similar proportion had minimal to mild and moderate to severe endometriosis. A total of 14.4% of the patients with endometriosis were smokers. Patients with endometriosis were equally diagnosed by either surgery or MRI (only when stage III–IV).

### 4.2. Comparison of miRNAs Expressed in Patients with and without Endometriosis

A total of 2633 miRNAs were found to be expressed in patients with endometriosis. The distribution of the miRNAs according to the AUC values is given in Appendix A. None had an AUC ≥ 0.70, and 2077 miRNAs had an AUC between ≥0.5 and <0.60. 

For the 57 miRNAs with an AUC ≥ 0.60, the sensitivity, specificity, accuracy, up/down regulation and AUC values are given in Appendix A. Of note, 5 miRNA were up-regulated in endometriosis patients (miR-6502-5p; miR-515-5p; miR-548j-5p; miR-29b-1-5p; miR-4748) and 2 miRNA were down regulated (miR-3137 and miR-3168). Eight members of the miRNA-548 family were identified (miR–548j–5p; miR–548p; miR–548ah–3p; miR–548l; miR–548q; miR–548f–5p; miR–548ay–3p; miR–548b–3p).

Among the 57 miRNAs with an AUC ≥0.6, 20 had not been reported before (miR–6502–5p; miR–548j–5p; miR–4748; miR–5697; miR–3124–5p; miR–4511; miR–3940–3p; miR–5009–5p; miR–10399; miR–3942–5p; miR–92b-5p; miR–4732–3p; miR–6789–5p; miR–6773–5p; miR–4466; miR–6802–5p; miR–4655–5p; miR–1343–5p; miR–8089; miR–3137), one had previously been reported in endometriosis (miR–124–3p), and the remaining 36 had previously been reported in benign and malignant disorders (miR–515–5p; miR–29b–1–5p; miR–548p; miR4999–5p; miR–6501–5p; miR–1270; miR–433–3p; miR–548ah–3p; miR–1278; miR548l; miR–1292–5p; miR–144–5p; miR–362–5p; miR–1285–3p; miR–3913–5p; miR–548q; miR–30e–3p; miR–151a–3p; miR–421; miR–27b–5p; miR–1910–3p; miR–542–5p; miR–548f–5p; miR–1250–5p; miR–1972; miR–548ay–3p; miR–6785–5p; miR–6777–5p; miR–4514; miR–4658; miR–1266–5p; miR–548b–3p; miR–6509–5p; miR–7107–5p; miR–6813–5p; miR–3168).

### 4.3. Relation between miRNA Expression and Signaling Pathways Known in Endometriosis

As mentioned above, the only miRNA previously reported in endometriosis was miR–124–3p (Appendix A). This miRNA is involved in ectopic endometrial cell proliferation and invasion by targeting ITGB3 and is downregulated by LncRNA-H19. It has also been described in other benign (peripheral arterial disease, hypertension, acute respiratory distress syndrome, Parkinson’s disease) and malignant (ovarian cancer, hepatocellular carcinoma, gastric cancer, glioma, breast cancer) disorders. The signaling pathways identified were mTOR, STAT3, PI3K/Akt, NF-κB, ERK, PLGF-ROS, FGF2-FGFR, MAPK, GSK3B/*β*–catenin. Besides its role in endometriosis, this miRNA is involved in cell proliferation, invasion, apoptosis, angiogenesis, inflammation, metastasis, and neurogenic functions. In addition, it has been associated with epithelial to mesenchymal transition, promoting trophoblast cell pyroptosis, chemosensitivity, and bone formation (Appendix A).

### 4.4. Relation between miRNA Expression and Signaling Pathways Involved in Disorders Other Than Endometriosis

The large majority of miRNAs differentially expressed in patients with and without endometriosis have not previously been identified as being involved in the pathophysiology of endometriosis. Most of them are known to be involved in numerous benign (atherosclerosis, diabetic nephropathy and retinopathy, renal and myocardial injury, vitiligo development, retinal degeneration, sickle cell disease, depressive disorders, epilepsy, early-onset preeclampsia, atrial fibrillation, hepatic steatosis, intracerebral hemorrhage, neurodegenerative disorders, bone formation, ovarian failure, nicotine initiation and addiction, endometrial receptivity in PCOS patients, response to estradiol, glaucoma), and malignant (hepatocellular carcinoma, retinoblastoma, prostate cancer, breast cancer, lung cancer, bladder cancer, thyroid cancer, osteosarcoma, ovarian cancer, gastric cancer, colorectal cancer, laryngeal squamous cell carcinoma, cholangiocarcinoma, chemo- and radiosensitivity) disorders, mainly in signaling pathways involved in key functions in carcinogenesis. The miRNA-associated disorders of the 10 miRNAs with the highest AUC and those differentially expressed in patients with and without endometriosis (up or down-regulated) are displayed Table 2, and in Appendix A for the others.

The main functions regulated by the miRNAs were: cell proliferation (22 miRNAs), apoptosis (16 miRNAs), adhesion/invasion (16 miRNAs), therapeutic (chemo- or radiotherapy), sensitivity (10 miRNAs), angiogenesis (5 miRNAs), immune response (5 miRNAs), inflammation (4 miRNAs), neurogenic function (2 miRNAs), related to pollutants (2 miRNAs), extracellular matrix remodeling or fibrosis (2 miRNAs), steroidogenesis or hormonal influence (1 miRNA), and other (18 miRNAs) (Table 3). The signaling pathways involved (Table 3) were: JAK/STAT (4 miRNAs), Notch1 (1 miRNA), FoxC1/Snail (2 miRNAs), Hippo (1 miRNA), NF–KB (4 miRNAs), YAP/TAZ (2 miRNAs), PIK3/Akt (7 miRNAs), HIF–1 alpha (2 miRNAs), JNK, Rap1b (1 miRNA), VEGF (1 miRNA), ERK (1 miRNA), PTH signaling (1 miRNA), Wnt/*β*-catenin (8 miRNAs), endogenous glucocorticoids (1 miRNA), insulin signaling pathway (1 miRNA), HBXIP (1 miRNA), GSK3B (1 miRNA), PTEN (1 miRNA), FOXO (3 miRNAs), MAPK (3 miRNAs), p53 (2 miRNAs), mTOR (2 miRNAs), TGF–ß (2 miRNAs). The main functions regulated by the 10 miRNAs with the highest AUC and those differentially expressed in patients with and without endometriosis (up or down-regulated) are displayed in Table 2 and Table 3, and in Appendix A for the others.

### 4.5. miRNA Expression Level According to Patient’s Characteristics

Among the 2633 reported miRNA, the top six according (miR-548j-5p, miR-29b-1-5p, miR-548p, miR-548l, miR-3913-5p, miR-124-3p) to their AUC value have been studied for their respective expression level based on the variation of rASRM stage, BMI, age, fertility status, smoking habit and hormonal treatment use. Figure 1, Figure 2, Figure 3, Figure 4, Figure 5 and Figure 6 display miRNA expression level. For those miRNA, we noticed in majority no significant variation of their expression level according to condition, excepted for miR-548l according to age factor (*p* = 0.03), miR-548p according to fertile status (*p* = 0.01), miR-29b-1-5p and miR-124-3p according to tobacco (*p* = 0.01 and *p* = 0.03 respectively) and miR-548p and miR-548l (*p* = 0.01 and *p* = 0.04) according to hormonal treatment use.

## 5. Discussion

Using miRNAome analysis, the present study contributes to establishing relations between miRNA expression in patients with endometriosis and various signaling pathways (often common to carcinogenesis) as well as identifying new potential pathways involved in the pathophysiology of endometriosis.

This is the first study to evaluate the expression of the currently known 2633 human miRNAs in endometriosis. However, it is important to note that all the human miRNAs were detected in the blood of patients with endometriosis. This poses the challenge of which miRNAs deserve to be analyzed to better understand the pathophysiology of endometriosis. Previous studies have demonstrated the pivotal role of miRNAs to improve diagnosis, prediction and forecasting for numerous diseases mainly based on micro-array of miRNAs or NGS sequencing using bioinformatics platforms with a limited number of miRNA, imposing a validation by qRT-PCR [26]. Another crucial issue is to evaluate whether results of NGS require qRT-PCR validation. Previous studies have shown that absolute NGS reads correlated modestly with qRT-PCR but fold changes, as used in the current study, correlated highly supporting that NGS is robust at relative but not absolute quantification of miRNA [85,86,87,88]. Moreover, as previously demonstrated, the number of microRNAs detected in biofluids by NGS and qRT-PCR was similar after filtering the data and applying thresholds supporting our results. In addition, recent studies validated the use of NGS technology to improve the diagnosing of using saliva RNA or to predict concussion duration and detect symptom recovery after mild traumatic brain injury. In this setting, in accordance with ’t Hoen et al. [89], bioinformatics allows the exhaustive analysis of all ARN fragments that are aligned and mapped, and their expression levels quantified, thus eliminating the need for sequence specific hybridization probes or qRT-PCR which are required in a microarray [26,90,91]. Moreover, NGS has the advantages of high sensitivity and resolution and excellent reproducibility but imposes considerable computational support [26,89,91]. So far, very few miRNAs (<300) have been used to determine the various biological mechanisms involved in the poorly understood and multifactorial pathophysiology of endometriosis. In the specific context of endometriosis, data from our prospective ENDOmiARN study [16] identified 57 miRNAs (2.2%) significantly expressed in patients with endometriosis with a diagnostic AUC of ≥0.60, suggesting a potential contribution in the pathophysiology of the disease. One crucial result of the current study was the identification of 20 miRNAs significantly expressed in patients with endometriosis but not previously reported in either benign or malignant disorders. This highlights the need for more fundamental research investigating their role in the pathophysiologic process of endometriosis. 

Among the 57 miRNAs differentially expressed in patients with endometriosis, only miR–124–3p has previously been reported in endometriosis. Interestingly, none of the miRNAs previously reported as potentially involved in endometriosis [10,92,93] were found to be a predictive marker of endometriosis in our cohort (i.e., with an AUC ≥0.6). Another crucial result is the high number of miRNAs not previously reported in the pathophysiology of endometriosis. Among these 57 miRNAs significantly expressed in patients with endometriosis, the vast majority have previously been detected in various cancers confirming that endometriosis shares several signaling pathways with carcinogenesis [94]. Overall, the miRNA–548 family seems to play a determinant role in endometriosis. Of the eight identified members of the miRNA-548 family, one has not been reported before (miR–548j–5p), three have been reported in carcinogenesis (miR-548p; miR-548l; miR-548b-3p) [66,67,68,69,70,71,72,95,96], and the remaining four (miR-548ah-3p; miR-548q; miR-548f-5p; miR-548ay-3p) are known to be involved in diabetic nephropathy, weight loss, circadian rhythm and smooth muscle contraction [97,98,99,100]. When analyzing the potential contribution of each miRNA in the pathophysiology of endometriosis, it is important to note that a single miRNA could be involved in multiple signaling pathways. The translational regulation by miRNAs involves intricately regulated composite interactions in which a single miRNA regulates the transcription of many mRNAs, and a single mRNA can be influenced by multiple miRNAs. The expression of miRNA in an individual is dynamic and is influenced by an array of factors, including age, ethnicity, the physiological stage of the body, the presence of various diseases, smoking, and various other external factors [93,94].

As mentioned above, the pathophysiology of endometriosis involves numerous signaling pathways which we cannot develop in a single report. In the present study, we did not perform experimental confirmation of the pathways potentially regulated by the miRNAs identified. Instead, we performed extensive literature research to set the basis for further well—designed works that will confirm the roles of the main miRNAs in the physiopathology of endometriosis. However, our work highlights the similar pathophysiological pathways involved in endometriosis and cancer genesis, with most miRNAs regulating cell proliferation (22 miRNAs), apoptosis (16 miRNAs), and adhesion/invasion (16 miRNAs). These three pathways could be determinant to promote endometriosis confirming preliminary analysis by Panir et al. [8]. These pathways compete with inflammation to promote endometriosis, although only four miRNAs have been directly linked to inflammation. Moreover, the contribution of angiogenesis and immune response has been underlined (5 miRNAs each). All these data reinforce the concept that endometriosis shares several signaling pathways of carcinogenesis.

Among the various pathways implicated in endometriosis, hypoxia plays a specific role in early phases of ectopic endometrial tissue survival induced by factor 1-α (HIF-1α) gene expression that is upregulated in endometriotic tissues [101]. Aberrant immune surveillance is thought to reduce the clearance of endometrial cells within the peritoneal cavity, permitting attachment, progression, and subsequent disease persistence [102,103]. The inflammatory mediators interleukin-1β (IL-1B) [104], TNF [105,106], and cyclooxygenase (COX)-2 [107] can be targeted by miRNAs in endometrial tissue. Lagana et al. reported the variation of the balance through the course of the disease between macrophages type 1 (pro-inflammatory) and 2 (pro fibrosis) that could be involved in the pathogenesis [108]. In our cohort, two identified miRNAs were associated with macrophages expression: miR-144 and miR-421. This latter was reported in the inflammatory process [109]. Aberrant estrogen and progesterone biosynthesis and metabolism contribute to the development of endometriosis [109] by increasing local estrogen production and promoting endometriosis development. Increased miR-142-3p seems to reduce steroid sulfatase and IL-6 activity, suggesting a dual effect on steroidogenic and inflammatory pathways in endometriosis [110]. Previous studies have shown a high expression of miR-210 increasing proliferation, angiogenesis, and resistance to apoptosis [111], whereas upregulation of miR-196 increases proliferation and anti-apoptotic mechanisms [112]. There are evidence that matrix metalloproteases (MMPs) also play a crucial role. Indeed, miR-520g acts on MMP2 synthesis that could act to enhance the degradation of the extracellular matrix and facilitate the anchoring of endometrial fragments in ectopic sites [113,114].

Although previous epidemiologic studies using Artificial Intelligence and Machine Learning have demonstrated the involvement of POPs in endometriosis, little data are available about their impact on miRNA expression in endometriosis. In the current study, we found two miRNAs significantly associated with pollutants: miR–421 (previously described in light pollution) and miR–542–5p (previously described in pulmonary fibrosis secondary to silicosis). Moreover, one miRNA (miR-548ay-3p) was previously found to regulate the circadian rhythm [100]. All these data underline the need to investigate other potential factors involved in the pathophysiology of endometriosis besides the classic POPs (octachlorodibenzofuran, cis-heptachlor epoxide, polychlorinated biphenyl 77, or trans-nonachlor) [3].

We focused our analysis on a few frequently observed specific pathways which may have a therapeutic impact, such as the Wnt/β-catenin, PI3K/AKT/mTOR, HOXA11, and Hippo pathways. To date, the main therapeutic options for patients with endometriosis are based on gonadotropin-releasing hormone agonists (as endometriosis is a well-known hormone-dependent pathology) or angiogenesis inhibitors but with inconsistent results [115,116,117,118,119,120]. The Wnt/β-catenin pathway: It has been demonstrated that Wnt signals are crucial for the activity of epithelial stem cells. The loss of the APC tumor suppressor gene function may lead to the deregulation of β-catenin stability [121]. Various targets activating or inhibiting Wnt signaling have been published (Porcupine, vacuolar ATPase, tankyrase Axin, PP2A, ARFGAP1 and GSK3) [122]. Moreover, soluble Wnt protein agonists have been shown to activate Wnt signaling in vivo [123] and several small molecule compounds (L807mts, Bio, CHIR, and SB-216763) [124] interfere with GSK3 and thus induce Wnt target gene expression. This could be of interest in the development of treatments for neurodegenerative disorders, including Alzheimer’s disease [125]. The Hippo pathway: The Hippo pathway has been shown to play a role in organ development, epithelial homeostasis, tissue regeneration, wound healing, immune modulation, as well as fibrosis that characterizes deep endometriosis [126]. Many of these roles are mediated by the transcriptional effectors YAP and TAZ [127,128]. The YAP/TAZ complex regulates pro-fibrotic factors and interferes with small-molecule inhibitors of PAI-1 and converges with pro-fibrotic signaling pathways such as TGFβ previously described in endometriosis [129]. Recently, verteporfin and VGLL4 mimetic peptides have been shown to inhibit YAP/TAZ-dependent transcription as well as suppress tumor growth. Thus, therapies targeting this transcription could potentially result in treatments for various diseases [130,131]. HOXA11: Long non-coding RNA HOXA11-AS has been shown to regulate target genes by epigenetic methylation and has been found to inhibit the Wnt signaling pathway via the upregulation of HOXA11, thus inhibiting proliferation and invasion properties [132]. Moreover, it has been found that overexpression of HOXA11-AS increases the membrane levels of CD44 [133] and decreases the expression of matrix metalloproteinase-2 [134], MMP-9, and vascular endothelial growth factors that are dysregulated in endometriosis. The PI3K/AKT/mTOR pathway: Previous studies have demonstrated that this pathway can modulate proliferation and angiogenesis in endometriosis [135], and that two rapamycin-analogues (temsirolimus and everolimus), already used in various cancers, inhibit mTOR signaling and reduce the growth of endometriosis implants [136].

Some limits of our study deserve to be underlined. First, in the present miRNAs analysis, we adopted as normalization thresholds read >1.5-fold as up-regulation and <0.5 downregulation which could be debated. However, this is in line which previous publications in the endometriosis fields [8,10,92]. Second, a surprising result of our work was the absence in the highly valuable miRNA of some miRNA previously reported as significant before [11,12,137,138,139,140,141]. This could be explained by the size of our cohort being much larger, which necessarily had an impact on the individual value of diagnostic performance for a single miRNA. Third, we focused on miRNAs with an accuracy ≥0.60, but it is not possible to rule out that some miRNAs with an accuracy between 0.50 and 0.59 might play a role in the pathophysiology of endometriosis. However, among the 2633 miRNAs detected in patients with endometriosis, 2077 had an AUC >b0.50 and <0.60. We would not have been able to comprehensively investigate all the signaling pathways potentially involved in endometriosis in a single report. Fourth, we only focused on miRNA expression, although several studies have demonstrated that other non-coding RNAs may play a role in the pathophysiology of endometriosis. Fifth, several reports underlined the potential impact of the menstrual cycle and the hormonal treatment on the miRNA expression, especially when endometrium samples have been analyzed [142,143]. In our cohort, a correlation was noted for two miRNAs (miR-548p and miR-548l) with the use of hormonal therapy which is conflicting with the results reported by the two aforementioned studies by Vanhie et al. and Moustafa et al. that found no impact of either treatment or menstrual phase on miRNAs levels. In addition, numerous other conditions may impact the expression level on miRNA, such as the endometriosis stage, age, BMI, tobacco and hormonal treatment use. Here, we reported the expression level for the six most accurate miRNA and demonstrated a low impact of such characteristics [10,92]. Sixth, we have not performed subgroup analysis to investigate the potential influence of miRNA expression in minimal cases to evolve toward severe endometriosis. This could be relevant, especially when further bench work will have deeply studied the miRNA identified as highly expressed in our cohort. Finally, due to the numerous pathways involved in the pathophysiology of endometriosis, we only focused on the most frequently observed, representing a true limit. However, all these data underline the need of further in vivo and in vitro analysis to confirm the pivotal role of miRNAs in endometriosis.

Our results provide evidence of the relation between miRNA profiles in patients with endometriosis and various signaling pathways implicated in its pathophysiology. In addition, the analysis of the miRNAome opens up new perspectives of investigation in the understanding of the underlying biological mechanisms involved not only in endometriosis but also in other pathologies qualified as multifactorial.

## Figures and Tables

**Figure 1 diagnostics-12-00175-f001:**
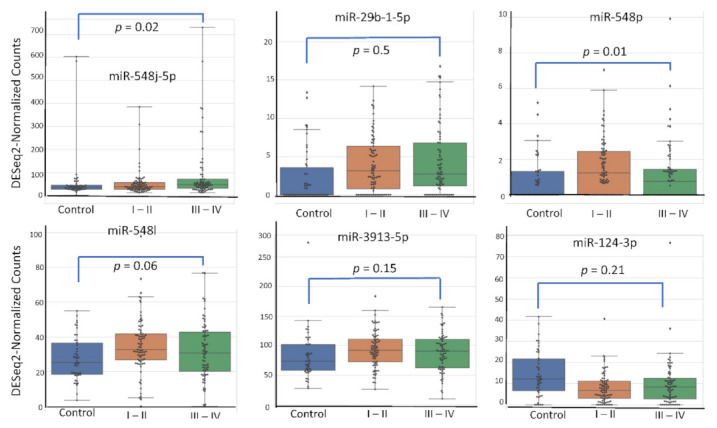
miRNA plasma expression according to endometriosis stage.

**Figure 2 diagnostics-12-00175-f002:**
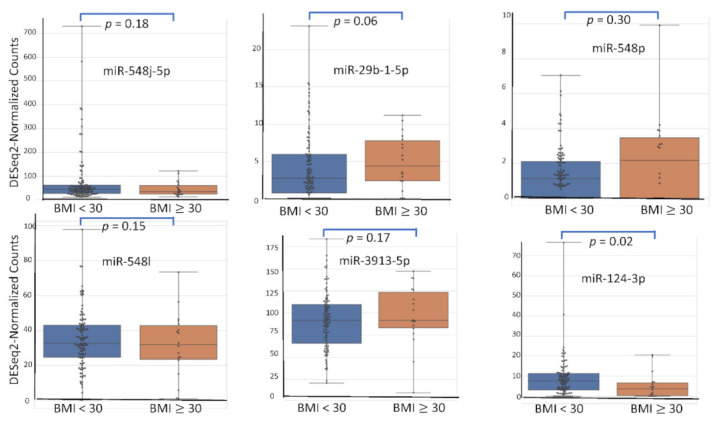
miRNA expression level according to BMI.

**Figure 3 diagnostics-12-00175-f003:**
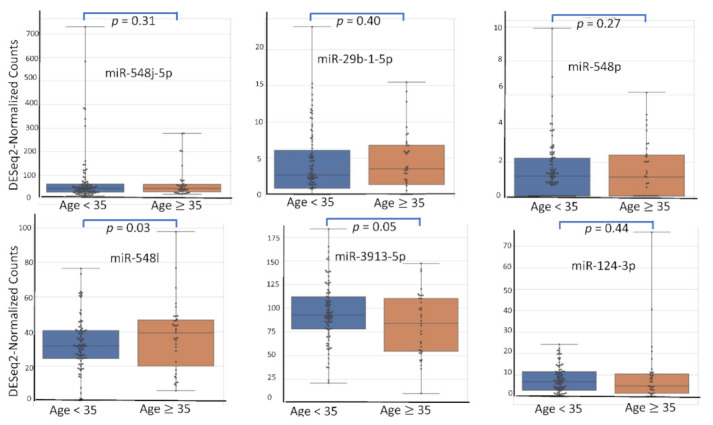
miRNA expression level according to Age.

**Figure 4 diagnostics-12-00175-f004:**
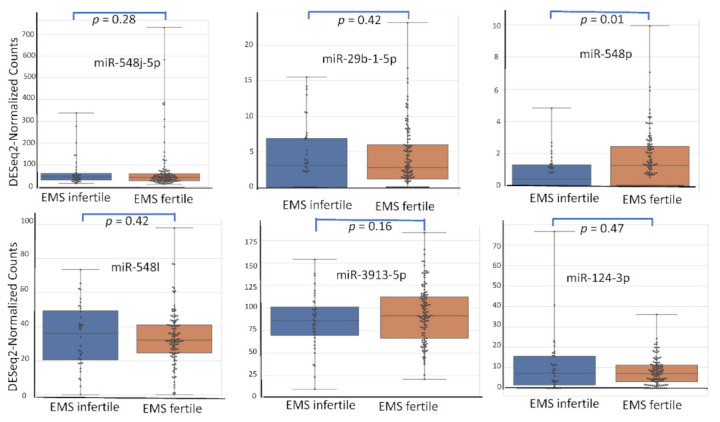
miRNA expression level according to fertility status.

**Figure 5 diagnostics-12-00175-f005:**
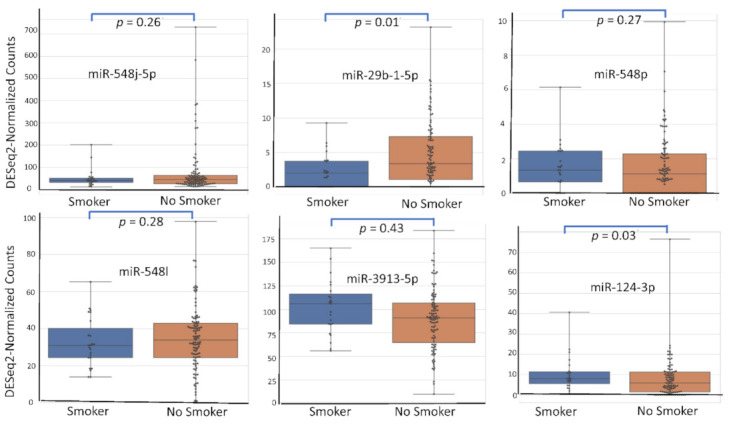
miRNA expression level according to tobacco.

**Figure 6 diagnostics-12-00175-f006:**
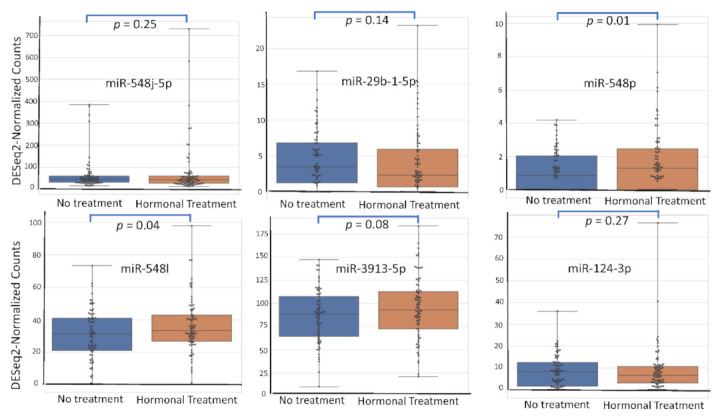
miRNA expression level according to hormonal treatment use.

**Table 1 diagnostics-12-00175-t001:** Demographic characteristics of the patients included in the ENDOmiRNA cohort.

	Controls *N* (%) *N* = 47	Endometriosis *N* (%) *N* = 153	*p*-Value
Age:mean (SD)	30.92 (13.79)	31.17 (10.78)	0.19
BMI: mean (SD)	24.84 (11.10)	24.36 (8.38)	0.53
Tobacco use	22 (14.4)	0 (0)	<0.01
rASRM classification - I–II - III–IV	- -	80 (52) 73 (48)	-
Control diagnoses - No abnormality - Leiomyoma - Cystadenoma - Teratoma - Other gynecological disorders	24 (51) 1 (2) 5 (11) 11 (23) 6 (13)	-	-
Dysmenorrhea	47 (100)	153 (100)	
Abdominal pain outside menstruation	21 (44)	89 (58.2)	0.70
Pain suggesting sciatica	10 (21)	70 (45.6)	0.02
Dyspareunia: mean (SD)	4.95 (3.52)	5.28 (3.95)	<0.01
Lower back pain outside menstruation	20 (42)	101 (66.0)	0.049
Painful defecation: mean (SD)	2.84 (2.76)	4.35 (3.47)	<0.01
Right shoulder pain near or during menstruation	3 (9)	26 (17.0)	0.22
Urinary pain during menstruation: mean (SD)	2.84 (2.76)	4.35(3.36)	<0.01
Blood in the stools during menstruation	4 (12)	30 (19.6)	0.24
Blood in urine during menstruation	8 (17)	21 (13.7)	0.42
Mode of diagnostic			-
*Surgery*	47 (100)	83 (54.2)	-
*Magnetic Resonance Imaging*	-	70 (45.8)	-

BMI: Body Mass Index; rASRM: revised American Society for Reproductive Medicine.

**Table 2 diagnostics-12-00175-t002:** miRNAs-associated benign and malignant disorders.

miRNAs	Up/Down Regulated	Benign Disorders	Malignant Disorders
miR-515-5p [36,37,38,39,40,41,42,43,44,45,46,47,48,49,50,51,52,53,54]	Up	Atherosclerosis	Hepato-cellular carcinoma, retinoblastoma, prostate cancer, Breast cancer, Lung cancer
miR-29b-1-5p [55,56,57,58,59,60,61,62,63,64,65]	Up	Helicobacter Pilori (Gastric cells), Spinal cord injury,	Breast cancer, Colon cancer, Oral squamous cell carcinoma, Bladder cancer
miR-548p [66,67,68,69]	-	-	Hepatitis B-mediated hepatocarcinoma
miR-548l [70,71,72]	-	Glaucoma	Hepatocellular carcinoma, Lung cancer
miR-3913-5p [73,74]	-	-	Lung cancer, Cholangiocarcinoma
miR-30e-3p [75,76,77,78,79,80,81]	-	-	Glioma, Hepatocellular carcinoma, ovarian cancer, colorectal cancer, clear cell renal cell carcinoma
miR-6813-5p [82]	-	-	Breast cancer
miR-3168 [83,84]	Down	Coronary atherosclerosis in patients with rheumatoid arthritis	-
miR-548j-5p	-	Never reported	Never reported
miR-6502-5p	Up	Never reported	Never reported
miR-4748	Up	Never reported	Never reported
miR-3137	Down	Never reported	Never reported

Link to Pollutants; Ster/Horm: Steroidogenesis or Hormonal influence; Therap Sens: Therapeutic sensitivity; EMT: Epithelium to Mesenchymal transition.

**Table 3 diagnostics-12-00175-t003:** miRNA-associated pathophysiologic pathways.

mirRNAs	Ad/ Inv	Prolif	Apopt	Angio	Inf	EMR	Met/Mig	Immune Resp/escT	Neuro f	LTP	Ster/Horm	Therap sens	Other
miR-29b-1-5p [55,56,57,58,59,60,61,62,63,64,65]	-	X	X	X	X	X	-	-	-	-	-	-	EMT
miR-548p [66,67,68,69]	X	X	X	-	-	-	X	-	-	-	-	X	Decreases Hepatic Apolipoprotein B Secretion and Lipid Synthesis
miR-548l [70,71,72]	X	-	-	-	-	-	X	-	-	-	-	-	-
miR-3913-5p [73,74]	-	-	-	-	-	-	-	-	-	-	-	X	-
miR-30e-3p [75,76,77,78,79,80,81]	-	X	X	-	X	-	-	-	X	-	-	-	Cardiomyocyte autophagy
miR-6813-5p [82]	-	-	X	-	-	-	-	-	-	-	-	-	-
miR-3168 [83,84]	-	-	-	-	-	-	-	-	-	-	-	-	-
miR-548j-5p	-	-	-	-	-	-	-	-	-	-	-	-	-
miR-6502-5p	-	-	-	-	-	-	-	-	-	-	-	-	-
miR-4748	-	-	-	-	-	-	-	-	-	-	-	-	-
miR-3137	-	-	-	-	-	-	-	-	-	-	-	-	-

Ad/Inv: Adhesion/Invasion, Prolif: Proliferation; Apopt: Apoptosis; Angio: Angiogenesis; Inf: Inflammation; EMR: Extracellular Matrix Remodeling; Met/Mig: Metastasis and Migration; Immune Resp/esc: Immune Response or escape; Neuro f: Neurogenic function; LTP. “-” is for “unreported or absent” and “X” is for “present”.

## Data Availability

The authors confirm that the data supporting the findings of this study are available within the article and its Appendix A.

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
