# Peer review of "Clues for Improving the Pathophysiology Knowledge for Endometriosis Using Plasma Micro-RNA Expression"

_diagnostics, 2022, doi:10.3390/diagnostics12010175_

Round 1
Reviewer 1 Report
I read with great interest the manuscript, which falls within the aim of this Journal. In my honest opinion, the topic is interesting enough to attract the readers’ attention. Nevertheless, authors should clarify some points and improve the discussion, as suggested below.
Authors should consider the following recommendations:
- Manuscript should be further revised in order to correct some typos and improve style.
- Accumulating evidence suggests that immune cells, adhesion molecules, extracellular matrix metalloproteinase and pro-inflammatory cytokines activate/alter peritoneal microenvironment, creating the conditions for differentiation, adhesion, proliferation and survival of ectopic endometrial cells. I would discuss these points in the light of new theories about the pathogenesis of endometriosis, referring to: PMID: 31663401; PMID: 28100109.
Author Response
We answered all comments in the common file along with the comments of Reviewer 2.
Thank you for these valuable comments.

Reviewer 2 Report
This article reported miRNA expression analyses in 200 patients with and without endometriosis. This discovery finding looks interesting and could give insight in understanding the pathogenesis, diagnosis and treatment of the disease. The large sample size analysed, and discovery of previously unreported miRNAs are strong side of this manuscript. However, this article has major concerns in presenting and interpretation of the data that needs to be addressed .
- The methodology needs additional detail explanation. Describe the normalization of the miRNA expression reads. In what basis do the author/s consider reads >1.5 fold over change as up-regulation and <0.5 down regulation. Describe the detail bioinformatics and software used in the analyses. Have the authors done signalling pathway analyses to map the miRNA functionality, or do they simply associate with previous report in the literature? It is not clear in the report, but I cannot see any pathway analyses report except matching with the reviewed articles. This not adequate for pathway analyses and to give concluding results the miRNA are mapped with the respective pathways. Proper experimental design and analyses are to reach into conclusion about the role of each miRNA the author reported.
- 2 More information is required on the demographic data and clinical finding of the patients including the age at menarche, parity, family history of endometriosis or endometrial/ovarian cancer, duration of presentation, method of diagnosis (confirmed by histology, confirmed by MRI) treatment received.
- How many of the patients with endometriosis have biopsy samples taken?
- Table 2 please revise the table format. It is distorted and separate the column and rows with lines. It needs to be summarized with reporting the most significant miRNA (the up-regulated and down regulated), the rest should be presented as supplementary data.
- Figures 1-6 all are not clear. Please increase the line thickness in both Y and x axes and increase the font size of the texts and numbers in both axes. Please include the P-value in the Figures to make clear for readers
- Why do you normalized the expression with small nuclear RNA gene U6. Justification needed. The normalization should be done with endogenous stably expressed miRNA. I know normalization of miRNA is very controversial and difficult decision, but the authors should have a clear stand on the normalization approach as this is very crucial in interpreting results. In this study, there were 47 patients without endometriosis one possible approach mechanism of normalization is the miRNA transcript reads from these samples.
- The up-regulated and down regulated miRNA in this report should require validation. There are two possible ways 1) using RT-qPCR which is a gold standard for quantification of miRNA 2) using in situ hybridization (RNAScope ISH) using the diagnostic tissue samples.
- In the aim authors stated to determine the sensitivity, specificity and functional of the miRNA which is very important concept and key of the study. However, there is no any single result that showed this concept. Please report it. I need to see at least functional data from in vitro experiment for a couple of the up-regulated and down regulated miRNAs.
- Minor correction in abstract line 42 and section 3.2 line 178 FGF2-EGFR pathway does not indicate logical signalling, FGF2 signals through FGFR, this needs revision
- Line 376 discussion conflicting ideas regarding hormone treatment, needs revision.
- Overall, the conclusion does not supported by the results.
- References are too much.
Author Response
All comments were answered in the attached file.
Reviewers 2
This article reported miRNA expression analyses in 200 patients with and without endometriosis. This discovery finding looks interesting and could give insight in understanding the pathogenesis, diagnosis and treatment of the disease. The large sample size analysed, and discovery of previously unreported miRNAs are strong side of this manuscript.
Thank you for this comment. We tried to do the best to improve knowledge on this topic
1. However, this article has major concerns in presenting and interpretation of the data that needs to be addressed.
We have considered and addressed all reviewer 2 comments to improve the quality of the manuscript. An important amount of data have been added, many discussion points have been raised in this revised version and overall this revised version is way more interesting than the original manuscript.
2. The methodology needs additional detail explanation. Describe the normalization of the miRNA expression reads. In what basis do the author/s consider reads >1.5 fold over change as up-regulation and <0.5 down regulation.
Thank you for this relevant comment. We agree with this comment which underlined for the normalization issue the lack of universal consensus between authors across the different publications.
If we considered the endometriosis field the most relevant studies from Panir et al (Human Reprod Updagte 2018), Moustafa et al. (American Journal of Obstetrics and Gynecology 2020), Vahnie et al. (Human Reprod, 2019), the thresholds used to define the normalization are (reads >1.5 fold as up-regulation and <0.5 down regulation).
To underline the importance of this comment, we have added the following sentence in the limits section.
“In the present mi RNA analysis, we adopted as normalization thresholds reads >1.5 fold as up-regulation and <0.5 down regulation which could be debated. However, this is in line which previous publication in the endometriosis fields...“
3. Describe the detail bioinformatics and software used in the analyses.
The process we used for bioinformatics analysis was the one previously described by Potla et al. (Osteoarthritis and Cartilage Open 2021). The revised version was enriched with a figure slightly adapted from their work that accurately described the process. This part has been added in the method section.
4. Have the authors done signalling pathway analyses to map the miRNA functionality, or do they simply associate with previous report in the literature? It is not clear in the report, but I cannot see any pathway analyses report except matching with the reviewed articles. This not adequate for pathway analyses and to give concluding results the miRNA are mapped with the respective pathways. Proper experimental design and analyses are to reach into conclusion about the role of each miRNA the author reported.
This comment is very relevant. Indeed, in this work, we associated miRNAs with previously reported pathway, which was a difficult and time-consuming work. Performing pathway analysis would have required selecting a limited number of miRNA which was not the intent of this work.
We decided to adopt the comment in the discussion section to underline this limit.
“In the present study, we did not perform experimental confirmation of the pathways potentially regulated by the miRNAs identified. Instead, we performed extensive literature research to set basis for further well – designed works that will confirm the roles of the main miRNAs in physiopathology of endometriosis.”
5. More information is required on the demographic data and clinical finding of the patients including the age at menarche, parity, family history of endometriosis or endometrial/ovarian cancer, duration of presentation, method of diagnosis (confirmed by histology, confirmed by MRI) treatment received.
As suggested, we added in the text and in the table more characteristics of the study population, including symptoms and mode of diagnostic. History of malignancy was one of the exclusion criteria of the ENDOmiARN cohort. The revised table was enriched a lot by this comment and provides valuable data for the reader to fully understand our work.
|
Controls N(%) N = 47 |
Endometriosis N(%) N = 153 |
P - value |
Age :mean (SD) |
30.92 (13.79) |
31.17 (10.78) |
0.19 |
BMI :mean (SD) |
24.84 (11.10) |
24.36 (8.38) |
0.53 |
Tobacco use |
22 (14.4) |
0 (0) |
< 0.01 |
rASRM classification - I-II - III-IV |
- - |
80 (52) 73 (48) |
- |
Control diagnoses - No abnormality - Leiomyoma - Cystadenoma - Teratoma - Other gynecological disorders |
24 (51) 1 (2) 5 (11) 11 (23) 6 (13) |
_ |
- |
Dysmenorrhea |
47 (100) |
153 (100) |
|
Abdominal pain outside menstruation |
21 (44) |
89 (58.2) |
0.70 |
Pain suggesting sciatica |
10 (21) |
70 (45.6) |
0.02 |
Dyspareunia : mean (SD) |
4.95 (3.52) |
5.28 (3.95)
|
<0.01 |
Lower back pain outside menstruation |
20 (42) |
101 (66.0) |
0.049 |
Painful defecation :mean (SD) |
2.84 (2.76) |
4.35 (3.47) |
<0.01 |
Right shoulder pain near or during menstruation |
3 (9) |
26 (17.0) |
0.22 |
Urinary pain during menstruation :mean (SD) |
2.84 (2.76) |
4.35(3.36) |
<0.01 |
Blood in the stools during menstruation |
4 (12) |
30 (19.6) |
0.24 |
Blood in urine during menstruation |
8 (17) |
21 (13.7) |
0.42 |
Mode of diagnostic |
|
|
- |
Surgery |
47 (100) |
83 (54.2) |
- |
Magnetic Resonance Imaging |
- |
70 (45.8) |
- |
Table 1: Demographic characteristics of the patients included in the ENDOmiRNA cohort
BMI: Body Mass Index
rASRM: revised American Society for Reproductive Medicine
6. How many of the patients with endometriosis have biopsy samples taken?
All patients undergoing diagnostic or operative laparoscopy underwent a systematic histological confirmation of endometriosis when potential lesions were present. This has been clarified in the text.
7. Table 2 please revise the table format. It is distorted and separate the column and rows with lines. It needs to be summarized with reporting the most significant miRNA (the up-regulated and down regulated), the rest should be presented as supplementary data.
Table 2 has been clarified according to the reviewer comment. The revised version now include a two-parts table 2: Table 2a solely refers to previously identified benign and malignant disorders associated with the miRNAs while Table 2b focuses on the pathways involved. This comment has helped improved the clarity of the manuscript, thanks.
8. Figures 1-6 all are not clear. Please increase the line thickness in both Y and x axes and increase the font size of the texts and numbers in both axes. Please include the P-value in the Figures to make clear for readers
The Figures 1 – 6 have been remastered to improve their quality and make them clear for the reader. Thanks for this comment.
9. Why do you normalized the expression with small nuclear RNA gene U6. Justification needed. The normalization should be done with endogenous stably expressed miRNA. I know normalization of miRNA is very controversial and difficult decision, but the authors should have a clear stand on the normalization approach as this is very crucial in interpreting results. In this study, there were 47 patients without endometriosis one possible approach mechanism of normalization is the miRNA transcript reads from these samples.
Thank you for your careful reading of our work. This is a typos error: normalization was performed using deSeq 2 and NOT using RNA gene U6, as it was clearly mentioned in our figures. We apologize for this typo.
10. The up-regulated and down regulated miRNA in this report should require validation. There are two possible ways 1) using RT-qPCR which is a gold standard for quantification of miRNA 2) using in situ hybridization (RNAScope ISH) using the diagnostic tissue samples.
We totally agree with this very relevant comment concerning the specific issue of normalization of miRNA expression. As aforementioned, we clarified the protocol used for normalization of miRNAs expression by adding the the miRNome Sequencing Analysis Pipeline process based on Potla. et al. As demonstrated by Haman et coll, various techniques are used to analyze miRNA expression. Some of them have both advantages and disadvantages (Table below). We opted to analyze the entire miRNome (more than 2600 human miRNAs known) using Learning Machine (LM) and Artificial Intelligence (AI) that are the most relevant biological techniques but imposing a hard bioinformatic analysis not imposing further validation by RT-qPCR or in situ hybridization. Some previous studies (Papari et al. DOI: 10.1016/j.fertnstert.2020.01.026) also used NGS technique but using a platform allowing only the analysis of a small number of miRNAs (most frequently 800 miRNAs) and without bioinformatic tool. Moreover, to our knowledge, none of previous studies in the specific setting of endometriosis used our protocol, explaining why a validation by RT-qPCR was required in these previous studies. However, our technique has also a disadvantage (Heinicke et al, RNA biology 2021). Although we used QIAseq that is the only kit that implements unique molecular identifiers (UMIs) during library preparation to reduce undesirable over-representation of miRNAs that were favorably amplified or sequenced, we did not use this technique. This might be considered as a limit of our study. Therefore, we added a sentence in the chapter on the limits of the current study; “ A potential bias linked to NGS technique, used in the present study, is the risk of over-representation of miRNAs that were favorably amplified or sequenced. Therefore, it should be interesting to evaluate the contribution of unique molecular identifiers (UMIs) to limit this specific risk”.
11. In the aim authors stated to determine the sensitivity, specificity and functional of the miRNA which is very important concept and key of the study. However, there is no any single result that showed this concept.
Indeed, these information are crucial. All these information are displayed in Annex 3 that goes along with the manuscript, to restrain the number of tables/figures within the text.
12. Please report it. I need to see at least functional data from in vitro experiment for a couple of the up-regulated and down regulated miRNAs.
As previously reported, as we used NGS technology, we could not report any in vitro experiment.
13. Minor correction in abstract line 42 and section 3.2 line 178 FGF2-EGFR pathway does not indicate logical signalling, FGF2 signals through FGFR, this needs revision
This typo has been revised accordingly.
14. Line 376 discussion conflicting ideas regarding hormone treatment, needs revision.
Indeed. Two miRNAs in our cohort were differentially expressed in patients using or not using hormonal treatment. The following sentences were added to the limit section of the revised version of the manuscript.
“In our cohort, a correlation was noted for two miRNAs (miR-548p and miR-548l) with the use of hormonal therapy which is conflicting with the results reported by the two aforementioned studies by Vanhie et al and Moustafa et al. that found no impact of either treatment or menstrual phase on miRNAs levels. Our work was not designed to assess formally this question which is a limit.”
15. Overall, the conclusion does not supported by the results.
We have considered the comment: our conclusion reflects both the extent and the limits of our work. Our results provide data of the relation between miRNA profiles in patients with endometriosis and various signaling pathways implicated in its pathophysiology. In addition, the analysis of the miRNAome opens up new perspectives of investigation in the understanding of the underlying biological mechanisms involved not only in endometriosis.
16. References are too much.
Indeed. As multiple references were added for each miRNA investigated, the list is very long. On the other hand, it reflects the extent of our literature’s research and it would be unfortunate to reduce the number of references used. We are willing to cut some references if it is determinant for the reviewer of course.
Again, thanks for the time spent and the valuable comments that greatly improved our work

Round 2
Reviewer 2 Report
First, I would like the authors for submitting the revised manuscript for review.
The revised manuscript has remarkable improvement in this current version. However, there are still sections that need to be addressed.
Figure 1-6 the numbers in the y-axis is very small and not visible. The font size should be increased and the line thickness in both axes should be thicker so the reader can easily understand the figures.
Table 2a and Table 2b need again improvement. Please report the most relevant (for those up or down regulated, particularly those presented in Figure 1. miRNA) and their pathological relation in other diseases and dysregulated pathways. The rest miRNAs with their respective references should be moved to supplementary data. This can help readers to focus on the most important miRNAs and if interested in the remaining they can refer to the supplementary data. This can also reduces the number of references in the main manuscript.
Remove Table 3 as the p-value has been included in the respective Figures and it is no longer important.
Validation data still required. Please report a validation (e.g using RT-qPCR) for the miRNAs reported in Figure 1.
There are still typo errors that need to be corrected e.g (25th percentile minus IQR or 75th) correct superscript
Author Response
All authors would sincerely thank the reviewer for the time spent again in reviewing our work and for the comments that greatly improved the value of our work. We hope this revised version will fully satisfy the reviewer.
RESPONSE TO REVIEWERS
Manuscript No: 1517123
Title: Clues for improving the pathophysiology knowledge for endometriosis using serum micro-RNA expression
Corresponding Author: Dr. Sofiane Bendifallah
By Yohann Dabi, Stéphane Suisse, Ludmila Jornea, Delphine Bouteiller, Cyril Touboul, Anne Puchar, Emile Daraï, Sofiane Bendifallah.
First of all, we would like to thank again the reviewer on his time spent reviewing again our work, and for his additional very constructive and relevant comments. All comments were considered carefully. A detailed response has been formalized for each of them in this document. We hope that these substantial changes, considering their relevance and their clinical impact, will find a favorable issue.
Reviewer
First, I would like the authors for submitting the revised manuscript for review.
The revised manuscript has remarkable improvement in this current version. However, there are still sections that need to be addressed.
Thank you for this very positive comment.
Review
- Figure 1-6 the numbers in the y-axis is very small and not visible. The font size should be increased and the line thickness in both axes should be thicker so the reader can easily understand the figures.
We consider each point of this comment (size, font, thickness) to improve de quality of the manuscript’s figures.
Please find, the figures which have been modified accordingly. Here is the example of Figure 1 in the revised version
Figure 1: miRNA serum expression according to endometriosis stage
- Table 2a and Table 2b need again improvement. Please report the most relevant (for those up or down regulated, particularly those presented in Figure 1. miRNA) and their pathological relation in other diseases and dysregulated pathways. The rest miRNAs with their respective references should be moved to supplementary data. This can help readers to focus on the most important miRNAs and if interested in the remaining they can refer to the supplementary data. This can also reduces the number of references in the main manuscript. Remove Table 3 as the p-value has been included in the respective Figures and it is no longer important.
We take carefully into account this comment. All the changes have been performed, only the top 10 miRNAs based on their AUC and those up or down regulated are displayed within main manuscript. The respective information concerning their regulation, dysregulated pathways and pathological relation are reported to improve the clarity of the manuscript.
Please find the final table 2.
mirRNAs |
Up / Down Regulated |
Benign disorders |
Malignant disorders |
miR-515-5p [36–54] |
Up |
Atherosclerosis |
Hepato-cellular carcinoma, retinoblastoma, prostate cancer, Breast cancer, Lung cancer |
miR-29b-1-5p [55–65] |
Up |
Helicobacter Pilori (Gastric cells), Spinal cord injury, |
Breast cancer, Colon cancer, Oral squamous cell carcinoma, Bladder cancer |
miR-548p [66–69] |
- |
- |
Hepatitis B mediated hepatocarcinoma |
miR-548l [70–72] |
- |
Glaucoma |
Hepatocellular carcinoma, Lung cancer |
miR-3913-5p [73,74] |
- |
- |
Lung cancer, Cholangiocarcinoma |
miR-30e-3p [75–81] |
- |
- |
Glioma, Hepatocellular carcinoma, ovarian cancer, colorectal cancer, clear cell renal cell carcinoma |
miR-6813-5p [82] |
- |
- |
Breast cancer |
miR-3168 [83,84] |
Down |
Coronary atherosclerosis in patients with rheumatoid arthritis |
- |
miR-548j-5p |
- |
Never reported |
Never reported |
miR-6502-5p |
Up |
Never reported |
Never reported |
miR-4748 |
Up |
Never reported |
Never reported |
miR-3137 |
Down |
Never reported |
Never reported |
The list of references has been consequently reduced by half. Table 2a is now table 2 and Table 2b is now Table 3. The previous Table 3 that summarized the p – values has been removed as required by the reviewer.
- Validation data still required. Please report a validation (e.g using RT-qPCR) for the miRNAs reported in Figure 1.
Thank you for this very challenging and debated question. Based on previous publication and to clarify this specific issue which is of major interest we suggest adding a few sentences in the discussion section:
“Previous studies have demonstrated the pivotal role of miRNAs to improve diagnosis, prediction and forecasting for numerous diseases mainly based on micro-array of miRNAs or NGS sequencing using bioinformatics platforms with a limited number of miRNA, imposing a validation by qRT-PCR [26]. In the present study, we opted to analyze the entire miRNAome using NGS and new generation of bioinformatics platforms avoiding a specific hybridization or qRT-PCR confirmation. Indeed, in accordance with A C ‘ t Hoen, et al. [85], bioinformatics allows the exhaustive analysis of all ARN fragments that are aligned and mapped, and their expression levels quantified, thus eliminating the need for sequence specific hybridization probes or qRT-PCR which are required in a microarray [26,86,87]. Moreover, NGS has the advantages of high sensitivity and resolution, and excellent reproducibility, but imposing considerable computational support [26,85,88].”
We reported few key sections from A C ’t Hoen et al., and Potla et al. concerning the application of NGS. The design of our study was made in accordance with those data.
“miRNAs can be sequenced using next-generation sequencing (NGS) platforms, in which after reverse transcription, millions of DNA fragments are sequenced in parallel. A range of platforms can be used for miRNA sequencing, including SOLiD (Applied Biosystems), Solexa, HiSeq, MiSeq, MiniSeq, NextSeq (Illumina), and Ion Torrent (Invitrogen), to name a few. Using bioinformatics, these fragments are aligned and mapped, and their expression levels are analysed, thus eliminating the need for sequence specific hybridisation probes which are required in a microarray. Moreover, NGS has the advantages of high sensitivity and resolution, and excellent reproducibility, though considerable computational support is required.
- A C ’t Hoen, P.A.C.; Ariyurek, Y.; Thygesen, H.H.; Vreugdenhil, E.; Vossen, R.H.A.M.; de Menezes, R.X.; Boer, J.M.; van Ommen, G.B.; den Dunnen, J.T. Deep sequencing-based expression analysis shows major advances in robustness, resolution and inter-lab portability over five microarray platforms. Nucleic Acids Res. 2008, 36, e141.
More recently, Potla et al. in 2021 stated that
“Next Generation Sequencing (NGS) technology has revolutionized the study of human genetic code, enabling a fast, reliable, and cost-effect method for reading the genome. Whereas “first generation” sequencing involved sequencing one molecule at a time, NGS involves sequencing multiple molecules in parallel.
- There are still typo errors that need to be corrected e.g (25th percentile minus IQR or 75th) correct superscript
This typo has been corrected and all the manuscript reviewed. Sorry for this.

Round 3
Reviewer 2 Report
I would like to thank again for resubmitting the revised version of the article for review. The manuscript has improved with this additional input.
I am not still happy with the figures' line thickness. It is highly desirable to present and sell scientific data in more clear and attractive way to readers. I modified one figure to be an example in the attached doc. It is good to look in other articles how data can be presented. Poor presentation signals poor data, although the quality of data is standard.
I understand the validation experiment is time and resource consuming but recommended to confirm a huge data through gold standard techniques.

Author Response
Review
I would like to thank again for resubmitting the revised version of the article for review. The manuscript has improved with this additional input.
I am not still happy with the figures' line thickness. It is highly desirable to present and sell scientific data in more clear and attractive way to readers. I modified one figure to be an example in the attached doc. It is good to look in other articles how data can be presented. Poor presentation signals poor data, although the quality of data is standard.
Please find, the figures which have been modified accordingly to the editor example. We thank the editor for the quality of its council.
I understand the validation experiment is time and resource consuming but recommended to confirm a huge data through gold standard techniques.
We agree with this comment which refers to a complementary validation (e.g using RT-qPCR).
While qRT-PCR is useful for quantifying the expression of a few genes or mi RNA, it can only detect known sequences. In contrast, RNA sequencing (RNA-Seq) using NGS can detect both known and novel transcripts. Because RNA-Seq does not require predesigned probes, the data sets are unbiased, allowing for hypothesis-free experimental design. For read-counting methods, such as gene expression profiling, the digital nature of NGS allows a virtually unlimited dynamic range. RNA-Seq quantifies individual sequence reads aligned to a reference sequence, producing absolute rather than relative expression values. This broad dynamic range enables detection of subtle changes in expression, down to 10%. Beyond quantifying gene expression, RNA-Seq can identify novel transcripts, alternatively spliced isoforms, splice sites, and small and noncoding RNA.1,2
- Ozsolak F, Milos PM. RNA-Sequencing: advances, challenges and opportunities. Nat Rev Genet.2011;12:87-98.
- Wang Z, Gerstein M, Snyder M. RNA-Seq: a revolutionary tool for transcriptomics.Nat Rev Genet. 2009;10:57-63.
As previously reported in our recent review we propose to add the following sentence to point out this important issue and improve the quality of the reported manuscript.
“Previous studies have demonstrated the pivotal role of miRNAs to improve diagnosis, prediction and forecasting for numerous diseases mainly based on micro-array of miRNAs or NGS sequencing using bioinformatics platforms with a limited number of miRNA, imposing a validation by qRT-PCR [26]. In the present study, we opted to analyze the entire miRNAome using NGS and new generation of bioinformatics platforms avoiding a specific hybridization or qRT-PCR confirmation. However, the validation experiment is still recommended to confirm a huge data through gold standard techniques. In this setting, A C ‘ t Hoen, et al. [85], reported that bioinformatics allows the exhaustive analysis of all ARN fragments that are aligned and mapped, and their expression levels quantified, thus eliminating the need for sequence specific hybridization probes or qRT-PCR which are required in a microarray [26,86,87]. Moreover, NGS has the advantages of high sensitivity and resolution, and excellent reproducibility, but imposing considerable computational support [26,85,88].
Again, thanks for the time spent.
